# Evaluation of CloudSat snowfall rate profiles by a comparison with in-situ micro rain radar observations in East Antarctica

Florentin Lemonnier[a,1] , Jean-Baptiste Madeleine[a], Chantal Claud[a], Christophe Genthon[a], Claudio Durán-Alarcón[b], Cyril Palerme[c], Alexis Berne[d], Niels Souverijns[e], Nicole van Lipzig[e], Irina V. Gorodetskaya[f], Tristan L'Ecuyer[g], and Norman Wood[g]

[a]Sorbonne Université, École normale supérieure, PSL Research University, École polytechnique, CNRS, Laboratoire de Météorologie dynamique, LMD/IPSL, F-75005 Paris, France
[b]Université Grenoble Alpes, CNRS, Institut des Géosciences de l'Environnement, Grenoble, France
[c]Development Centre for Weather Forecasting, Norwegian Meteorological Institute, Oslo, Norway
[d]Environmental Remote Sensing Laboratory, Environmental Engineering Institute, School of Architecture, Civil and Environmental Engineering, École Polytechnique Fédérale de Lausanne, Lausanne, Switzerland
[e]Department of Earth and Environmental Sciences, KU Leuven – University of Leuven, Heverlee, Belgium
[f]Department of Atmospheric and Oceanic Sciences, University of Wisconsin-Madison, Madison, Wisconsin, USA
[g]Centre for Environmental and Marine Studies, Department of Physics, University of Aveiro, Portugal

**Correspondence:** Florentin Lemonnier (flemonnier@lmd.jussieu.fr)

**Abstract.**

The Antarctic continent is a vast desert, the coldest and the most unknown area on Earth. It contains the Antarctic ice sheet, the largest continental water reservoir on Earth that could be affected by the current global warming, leading to sea level rise. The only significant supply of ice is through precipitation, which can be observed from the surface and from space. Remote sensing observations of the coastal regions and the inner continent using CloudSat radar give an estimated rate of snowfall but with uncertainties twice as large as each single measured value, whereas climate models give a range from half to twice the space-time averaged observations. The aim of this study is the evaluation of the vertical precipitation rate profiles of CloudSat radar by comparison with two surface-based Micro-Rain Radars (MRR), located at the coastal French Dumont d'Urville station and at the Belgian Princess Elisabeth station, located in the Dronning Maud Land escarpment zone. This in turn leads to a better understanding and reassessment of CloudSat uncertainties. We compared a total of four precipitation events, two per station, when CloudSat overpassed within 10 km of the stations and we compared these two different datasets at each vertical level. The correlation between both datasets is near-perfect, even though climatic and geographic conditions are different for the two

stations. Using different CloudSat and MRR vertical levels, we obtain 10km space-scale and short time-scale (a few seconds) CloudSat uncertainties from -13 % up to +22 %. This confirms the robustness of the CloudSat retrievals of snowfall over Antarctica above the blind zone and justifies further analyses of this dataset.

**1 Introduction**

In the context of global warming, predicting the evolution of the Antarctic ice sheet is a major challenge. Snowfall is the main input of the ice sheet mass balance, but it is difficult to estimate its amount. Indeed precipitation characteristics depend on the region of Antarctica. In coastal areas, precipitation is influenced by cyclones and fronts (Bromwich, 1988) and a few times a year, these fronts intrude on the high continental plateau, likely bringing most of the snow accumulation (Genthon et al., 2016),

the remaining annual precipitation rate being in the form of "Diamond Dust" (thin ice crystals) under clear-sky conditions (Bromwich, 1988; Fujita and Abe, 2006).

Some field campaigns with in-situ observations were conducted to estimate local snow accumulations (Arthern et al., 2006; Eisen et al., 2008), but ground-based measurements are difficult in Antarctica and the size of this continent (twice the size of Australia) does not permit one to cover and study the whole occurrence, rate and distribution of precipitation. Moreover, accu-

mulation observed from stake measurements is a poor proxy for snowfall as it is strongly affected by local winds (Souverijns et al., 2018a).

CloudSat and its cloud-profiling radar (CPR) provide the first real opportunity to estimate the precipitation at polar continental scale (Stephens and Ellis, 2008; Liu et al., 2008). Since August 2006 CloudSat has been observing solid precipitation through the atmosphere, which leads to the first multi-year, model-independent climatology of Antarctic precipitation (Palerme

et al., 2014). Using two CloudSat products to determine the frequency and the phase of precipitation and its rate, Palerme et al. (2014) established a mean snowfall rate from August 2006 to April 2011 of 171 mm.w.e/year over the Antarctic ice sheet, north of $82^o$S. Palerme et al. (2018) recently revisited the data and reduced this estimate to 160 mm.w.e/year. It is worth noting that this rate is given at an altitude of about 1200 m above ground level (m.a.g.l.) due to the reflectivity of snow interfering with radar waves near the surface (the so-called ground clutter, Kulie and Bennartz (2009); it is worth noting that close to the coastal

areas and over the ocean, this vertical limit for observation can be lower). Boening et al. (2012) showed that there is a good agreement between CloudSat and ERA-Interim precipitation over Dronning Maud Land, responsible for the total ice sheet mass anomalies detected by GRACE, but currently the estimated uncertainties for the satellite snowfall rate range between 50 % and 175 % (Wood, 2011). Palerme et al. (2017) showed that ERA interim is also in good agreement with CloudSat at the continental scale.

In January 2010, a first micro rain radar (MRR) used for precipitation studies was installed in Antarctica at the Belgian Princess Elisabeth station in the escarpment zone of Dronning Maud Land (PE, 71$^o$57'S,23$^o$21'E at 1392 above mean sea level) in the context of the Belgian project HYDRANT (The Atmospheric branch of the HYDRological cycle in ANTarctica) (Gorodetskaya et al., 2015). The PE station is located in the escarpment zone of Dronning Maud Land with Sør Rondande

mountains to the south of it (for detailed description of the station meteorological conditions see Gorodetskaya et al. (2013) and Souverijns et al. (2018a)). In November 2015, in the context of the French-Swiss APRES3 project (Antarctic Precipitation, Remote Sensing from Surface and Space) new instruments were deployed at the French station Dumont d'Urville on the coast of Adélie Land, in East Antarctica (DDU, 66$^o$40'S, 140$^o$00'E at 42 a.m.s.l.) leading to unprecedented weather radar observations of precipitation by a scanning X-band polarimetric radar and a K-band vertically profiling micro-rain radar (Grazioli

et al., 2017a). A comparison of MRR and CloudSat derived surface snowfall product showed that CloudSat is able to accurately represent the snowfall climatology with biases smaller than 15%, outperforming ERA-Interim (Souverijns et al., 2018b). Moreover, CloudSat's blind zone (lowest measurement available at about 1200 m above the surface) leads to surface precipitation amounts being underestimated by about 10 % on average although differences during specific events can be much larger (Maahn et al., 2014). This paper focuses on the vertical structure of precipitation.

With the aim of improving CloudSat radar uncertainty estimates using ground-based observations, CloudSat snowfall retrievals over Dumont d'Urville and Princess Elisabeth stations were compared with MRR data on a total of 4 concurrently recorded snowfall events. During the MRR observing periods, there were 14 overflights over DDU and 63 over PE. These overflights are short, typically a few seconds, explaining why we actually detected snow for only 4 of them. According to these events and using the deviation of CloudSat precipitation rates from MRR observations, its uncertainties were reassessed.

A systematic difference is found between CloudSat and the ground radars, by comparing their very low snowfall rates. This difference could be due to limitations in sensitivity or attenuation of the MRRs.

As a first step, we characterize the general weather conditions of the four cases (section 3.1). Then, a comparison is done between CloudSat and the vertical MRRs precipitation profiles (section 4.1 and 4.2). From this comparison we highlight a systematic difference (section 4.3), then from a statistical study described in Appendix A, a nearly-perfect correlation between

MRR and CloudSat datasets is derived (section 4.4). To conclude, we assess a new range of CloudSat uncertainties at short time scale (a few seconds) and 10km space scale (section 4.4).

## 2    Methods

### 2.1    CloudSat cloud-profiling radar

The CloudSat cloud-profiling radar is a nadir-looking 94 GHz radar which measures the signal backscattered by hydrometeors.

Radar reflectivity profiles are divided into 150 vertical bins with a resolution of 240 m, with a 1.7 x 1.3 km$^2$ footprint and up to 82$^o$ of latitude. CloudSat has been operating full-time since April 2006, but because of a dysfunctional on-board battery, has been only able to provide daylight observations since April 2011. The satellite is characterized by a period of 16 days, so it exactly overpasses a location every 16 days. DDU is overpassed by a descending orbit whereas PE is overpassed by ascending

and descending orbits which are less than 10 km away from each station. The CloudSat vertical bins are relative to the geoid and depending on the altitude where the stations are located, the first exploitable bin (out of the ground clutter alteration altitude) varies significantly. Moreover, in locations where the ice does not interfere much with the radar signal (ocean and some coastal areas), the ground clutter layer is thinner and lower altitude bins can be used. We are using at DDU CloudSat profiles from the

4th bin, which is located at 961 m.a.g.l. At PE the first exploitable bin is the 5th, which is located at 1043 m.a.g.l. We use the 2C-SNOW-PROFILE product (Wood, 2011) which retrieves profiles of liquid-equivalent snowfall rates. The product is based on assumptions on snow particle size distribution, micro-physical and scattering properties which induce many uncertainties in the calculation of the relationship between radar reflectivity and snowfall rate (see section 2.2).

## 2.2   Micro rain radars

The MRR is a vertically profiling Doppler radar operating at a frequency of 24.3 GHz (K-band) with a beamwidth of $2°$ (around 50 m in diameter at a 3000 m altitude). At both stations, the resolution was set to 100 m per bin ranging from 300 m – first valid available measurements – to 3000 m. However, we only consider the data up to 2500 m because of the change in the snow microphysical properties above this altitude (Grazioli et al., 2017a). The MRR's raw measurement – Doppler spectral densities – is available at 10s temporal resolution. The collected data were processed using the IMProTool developed by

(Maahn and Kollias, 2012). At DDU, the radar reflectivity derived from MRR was calibrated by comparison with a colocated X-band polarimetric radar over the period from December 2015 to January 2016 (for more details, see Grazioli et al. (2017a)). Through this calibration with the second radar, the reflectivity (at X-band) is converted into snowfall rates using a $Z_e/S_r$ relation (Grazioli et al., 2017a) :

$$Z_e = 76 * S_r^{0.91} \tag{1}$$

with $Z_e$ the radar reflectivity (in dBZ) and $S_r$ the snowfall rate (in mm/h). Grazioli et al. (2017a), proposed a range of values of [69-83] for the prefactor and [0.78-1.09] for the exponent corresponding to a confidence interval of 95 %.

For the instrument operating at PE station, hereafter called MRR2, the average Ze/Sr relation is given by Souverijns et al. (2017) :

$$Z_e = 18 * S_r^{1.10} \tag{2}$$

The range of prefactor [11-43] and exponent [0.97-1.17] for this equation spans a confidence interval of 40 % due to the summation of uncertainties in particle size, shape, measurement and conversion from reflectivity $Z_e$ to snowfall rate $S_r$. For this study, the used MRR2 data are processed with the Maahn and Kollias (2012) algorithm. Unlike Souverijns et al. (2017), we did not calibrate the ground radar dataset with CloudSat reflectivities because (1) we want an independent evaluation of the CloudSat CPR dataset, and (2) we do not consider surface precipitation rate comparisons. The mean precipitation profiles

obtained over the MRR observation periods (2015-2016 for DDU and 2012 for PE) were also used to evaluate how typical

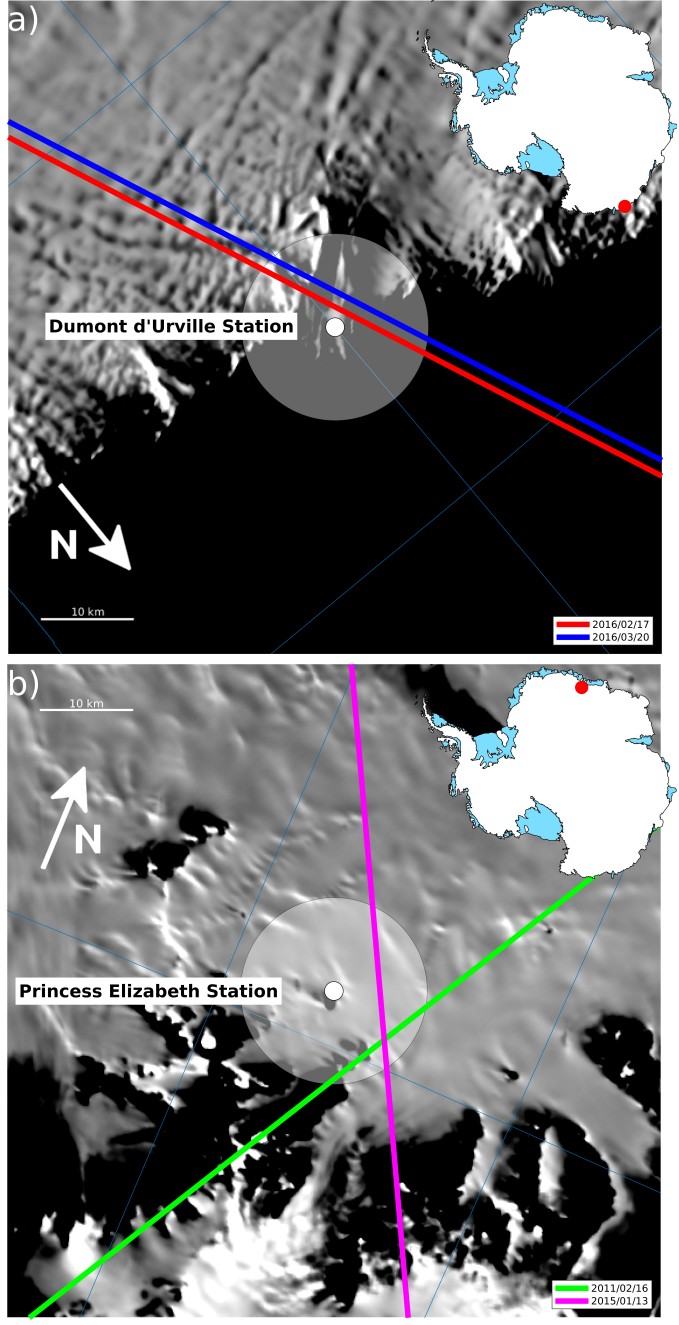

**Figure 1. a)** CloudSat radar tracks passing over the French Dumont d'Urville station (DDU) in red for the 17 Feb 2016 and in blue for the 20 March 2016. **b)** CloudSat radar tracks passing over the Belgian Princess Elisabeth station (PE) in green for the 16 Feb 2011 and in magenta for the 13 Jan 2015. We only considered the measured profiles passing within a 10 km radius represented by a white disc around the stations. The background image is the hill shaded topography obtained with MODIS MOA2004 (Haran et al., 2005)

.

the 4 precipitation events are (Durán-Alarcón et al., 2019). They are obtained using the same Ze/Sr relationships as the ones introduced earlier (see equations (1) and (2)) and are separated into quantiles. According to Maahn and Kollias (2012), the minimum detection of both MRR varies between -14 and -8 dBZ, corresponding to 0.00122 – 0.00546 mm/h at DDU and 0.00385 – 0.0135 mm/h at PE. However these values correspond to theoretical cases of clear sky. Therefore we analyzed the density probability functions of the MRR1 at 3 different levels to determine a minimum threshold of detectability of ground radars (figure 6 in Appendix). We used the lowest level out of the ground clutter layer (about 1200 m.a.g.l.) and selected a threshold of 0.005 mm/h (see the vertical dashed line in figure 6 in Appendix).

## 2.3 Radiosondes

A radiosonde is a meteorological device containing a set of sensors to measure the characteristics of the atmosphere from ground level to an altitude ranging from 25 up to 30 km. Parameters measured are temperature, relative humidity, wind speed, wind direction, pressure.

At DDU, the used radiosonde system is a METEOMODEM M10. The relative humidity accuracy is $3\%$ and its temporal resolution is 2 s. The temperature measurement is realized every 1s with an accuracy of 0.3°C. At PE, the ground receiving system used are GRAW-GS-E and GRAW radiosondes DFM-09-QRE. Relative humidity is measured with an accuracy of $3\%$ and a temporal resolution of 4 s. The accuracy and the temporal resolution of the temperature measurements are 0.2°C and 3-4 s.

## 3 Meteorological conditions of the four recorded snowfall events

### 3.1 Event characteristics

We summarize in table 1 the characteristics of the four recorded precipitation cases, when both CloudSat and ground-based MRRs simultaneously record a snowfall event, and when the satellite is in the vicinity of the stations. Due to the CloudSat delay of revisit, satellite overflights near the DDU station are located either less than 10 km and then more than 80 km away. CloudSat tracks passing through a radius of 10 km around each station (figure 1) were selected. Each CloudSat flyby over a station takes less than 10 sec and covers a distance between 11.90 km and 17.33 km. We consider that the four associated weather systems are static in regards with CloudSat satellite overfly. However, MRRs are stationary and local precipitation patterns are typically associated with transient large- and meso-scale weather systems. We therefore analyzed the synoptic conditions by using radiosonde data and reanalysis (ERA-Interim) from the European Centre for Medium-Range Weather Forecasts (ECMWF) in order to determine the adequate MRR time-series corresponding to CloudSat observations. We estimated a duration for which MRR observing conditions agree most with those of CloudSat using the following equation :

$$\Delta t_{avg} = \frac{\Delta x_{sat}}{V_{wind}} \tag{3}$$

**Table 1.** Weather conditions and instrumental characteristics for DDU and PE stations. Wind velocity is vertically averaged over the first 3 km of the atmosphere. Times are converted from UTC and displayed in Local Time (LT), DDU is UTC+10 and PE is UTC+03. Symbol * denotes that weather conditions were retrieved from ERA-I profiles, instead of a radiosonde.

| | Dumont d'Urville | | Princess Elisabeth | |
|---|---|---|---|---|
| | 2016/02/17 | 2016/03/20 | 2011/02/16 | 2015/01/13 |
| Wind averaged velocity (km/h) | 22.84 | 25.05 | 18.85 | 32.48 |
| CloudSat track length (km) | 17.33 | 15.16 | 11.90 | 16.23 |
| Start time of CloudSat obs. (LT) | 15:44:14 | 15:44:24 | 01:53:48 | 16:42:37 |
| End time of CloudSat obs. (LT) | 15:44:43 | 15:44:53 | 01:53:50 | 16:42:41 |
| Start time of MRR obs. (LT) | 15:21:00 | 15:26:00 | 01:34:00 | 16:26:00 |
| End time of MRR obs. (LT) | 16:07:00 | 16:02:00 | 02:12:00 | 17:00:00 |
| Radiosounding time (LT) | 10:00:00 | 10:00:00 | 03:00:00* | 13:58:00 |

where $\Delta t_{avg}$ represents the temporal range of the MRR observations wrapping CloudSat overflight date, $\Delta x_{sat}$ is the length of the track inside the 10 km radius area over stations and $V_{wind}$ is the vertically averaged wind velocity. All characteristics are shown in table 1.

### 3.1.1 Events at DDU

The February $17^{th}$ 2016 precipitation event at DDU was overflown by CloudSat in the local afternoon. It occurred on the edge of a low pressure system which was approaching the station, in agreement with the radiosounding launched in the morning at 09:00 LT. Indeed on figure 2b, 2c, above 1.5 km, a westerly wind brings moisture and a warmer air mass. The radiosounding also shows wind with a continental origin below 1 km which brings a relatively dry air. The recorded precipitation profile (figure 3a) presents a low-level sublimation below 1 km and thus suggests that this layer might be dried by continental winds, according to wind direction, relative humidity and temperature profiles.

Located between two low pressure systems, the March $20^{th}$ 2016 radiosounding is characterized by a shear between continental and oceanic winds below 500 m, marked by an inversion of relative humidity (figure 2e, figure 2f). Being at the rear margin of the first passing low pressure system, it explains the easterly origin of oceanic winds. It is followed by a strong event recorded in the afternoon by the radars, with katabatic winds blowing down the ice cap and sublimating precipitation at low altitude below 1000 m (figure 3b). This kind of dry air leading to significant low-level sublimation of snowfall is well documented by Grazioli et al. (2017b).

### 3.1.2 Events at PE

To analyze the vertical meteorological profiles at the Princess Elisabeth station we used ERA-Interim reanalysis, due to the absence of air-sounding campaign during the third precipitation event period. The February $15^{th}$ 2011 precipitation night

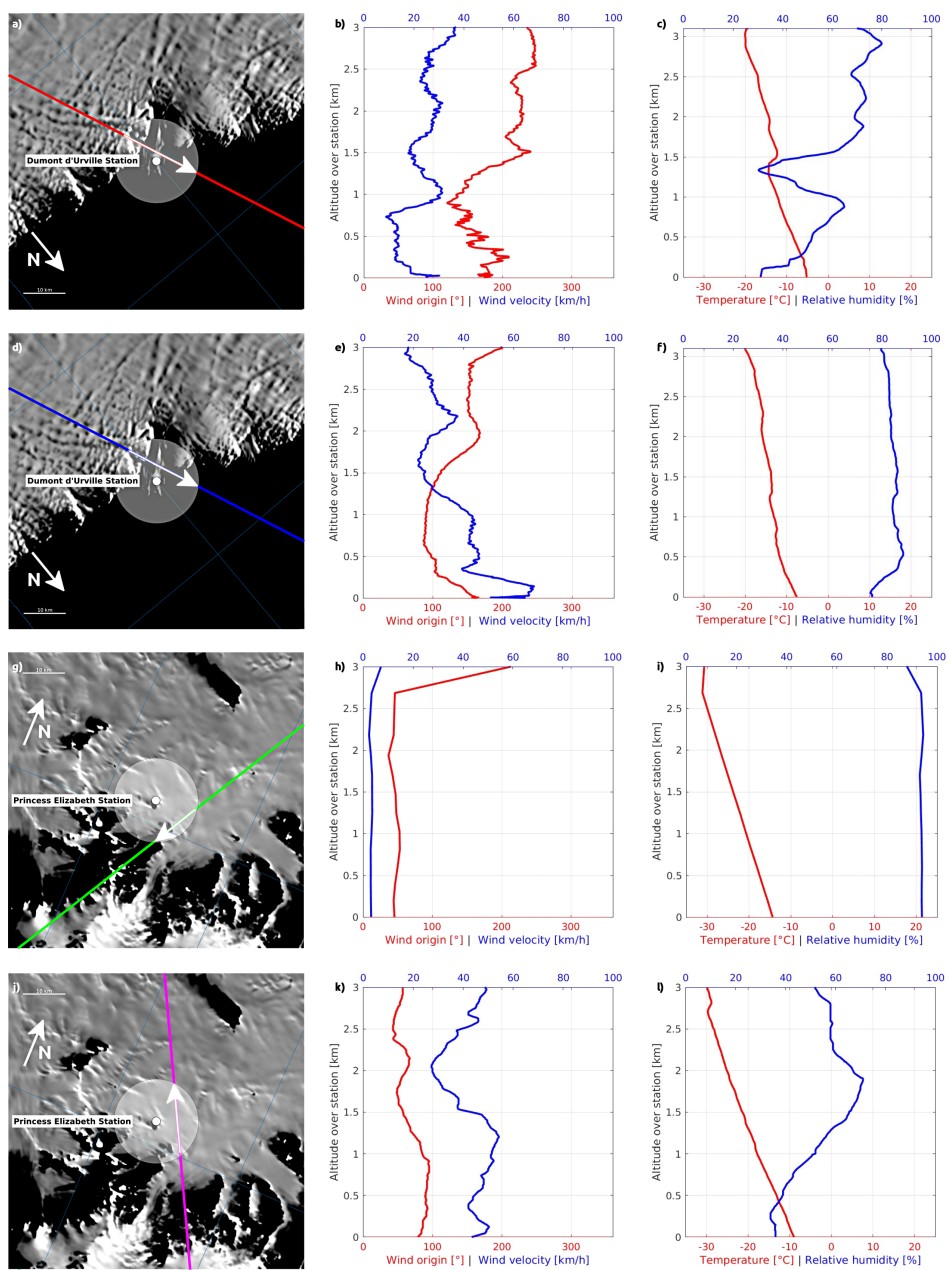

**Figure 2.** Vertical profiles of the lower tropospheric meteorological parameters over DDU and PE stations for the four precipitation events. The radiosonde launch times are summarized in table 1. **a-d-g-j)** First column shows each station location, selected CloudSat tracks and their directions. The white disk represents a 10 km-radius area around each station where we consider the CloudSat measurements. First row : February $17^{th}$ 2016; second row : March $20^{th}$ 2016; third row : February $15^{th}$ 2011; fourth row : January $13^{th}$ 2015. The background image is the hill shaded topography obtained with MODIS MOA2004 (Haran et al., 2005) **b-e-h-k)** The second column shows wind velocities (blue solid line) and wind directions ($0^o$ indicating from the North) (red solid line) over the stations gathered with radiosoundings except for the **h)** plot, which is obtained with ERA-Interim. **c-f-i-l)** The third column shows air temperatures (red solid line) and relative humidities with respect to ice (blue solid line) over the station obtained with radiosoundings, except for the **i)**, deduced from ERA-Interim.

event is characterized by a large low pressure system north-west of the PE station blocked by a high-pressure ridge to the east directing a strong moisture flux defined as an atmospheric river directly to PE station. It is a significant snowfall event that caused anomalous increase in Dronning Maud Land surface mass balance (Gorodetskaya et al., 2014). The westerly origin of the high-altitude wind observed in figure 2 is dominated by the circumpolar atmospheric circulation. At the resolution of the reanalysis ($0.75^o$ in longitude and latitude), it is difficult to observe any orographic impact on the weather around the Princess Elisabeth station.

The fourth observed radiosounding, released 3 hours before the January $13^{th}$ 2015 event in the afternoon, is explained by a low pressure system located north-west of PE and a strong, constant in altitude, easterly wind (figure 2k). The temperature and relative humidity suggest a cloudy weather with a dryer and hotter boundary layer (figure 2l). The observed precipitation profile suggests in-cloud snowfall and virga (figure 3d). This is confirmed with a backscatter profile measured by a ceilometer installed at PE station (see figure 5 in appendix) observing a passing cloud over the station during the record of the precipitation event by the CloudSat and MRR radars.

### 3.2 Estimation of the confidence in CloudSat reports

All CloudSat measurements were selected within a 10 km-radius from each station and averaged for each vertical bin. A variance on the CloudSat retrievals is computed for the duration of each overpass (see figure 7 in appendix).

The MRR confidence intervals are calculated using the range of Ze-Sr parameters given by Grazioli et al. (2017a) for the Dumont d'Urville station and Souverijns et al. (2017) for the Princess Elisabeth station. At DDU, according to Grazioli et al. (2017a), for an altitude higher than 2500 m where there is a crystal dominance for precipitation, the used parametrizations for Ze-Sr conversion are not adapted anymore. That is why MRR measurements are considered and compared to equivalent CloudSat vertical bins only in the first 2500 m of the atmosphere. In contrast with the coastal areas, we would expect less riming at PE compared to DDU, while aggregates are expected to occur at PE given the measured large particle sizes (Souverijns et al., 2017). Also the low variability in the vertical profile of mean Doppler vertical velocity at PE suggests that aggregation/riming of particles is not frequent in this region and hydrometeor type is relatively constant in the vertical profile (Durán-Alarcón et al., 2019). Without this change in the proportion of the different hydro-meteors, the ground-based Ze-Sr relationships would be still valid higher up.

## 4 Results and discussion

### 4.1 Precipitation profiles

Focusing on the Dumont d'Urville station, figure 3a shows a good agreement between CloudSat and MRRs snowfall rates for each vertical level. Indeed, averaged satellite precipitation rate at all levels is included within the 95 % MRR confidence interval. The MRR profile presents a maximum of the snowfall rate of 0.75 mm/h at 750 m and an inversion of the precipitation rate likely due to low-level sublimation processes, whereas the ground clutter prevents CloudSat from seeing the inversion.

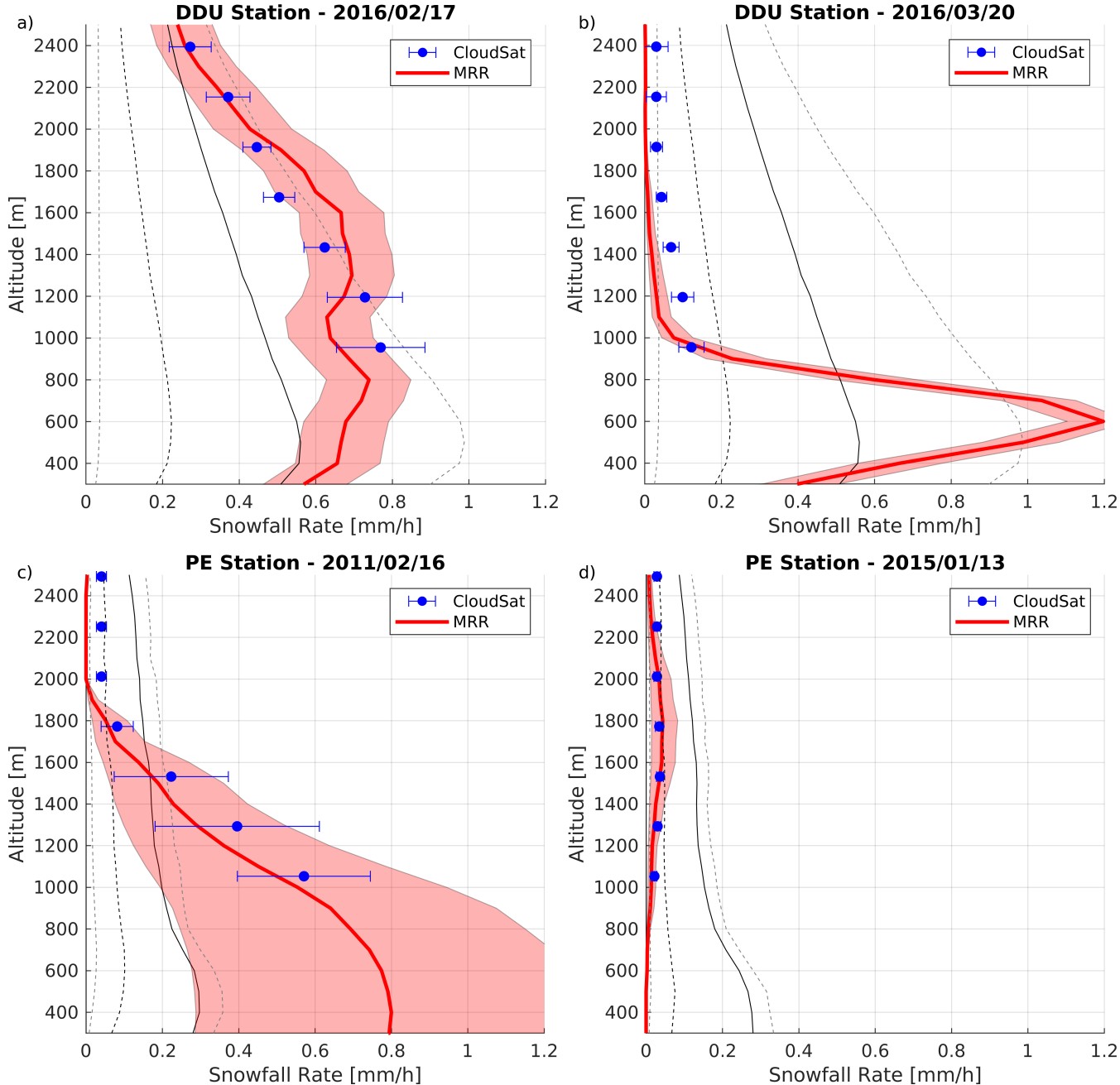

**Figure 3. a)** Comparison between CloudSat (blue dot with 2-$\sigma$ standard deviation bars) and MRR (red solid line with shaded area representing 95% confidence interval) for the February 17[th] 2016 precipitation event at DDU. **b)** Same as **a** for the March 20[th] 2016 event at DDU. **c)** Comparison between CloudSat (blue dot with 2-$\sigma$ standard deviation bars) and MRR (red solid line with shaded area representing 40% confidence interval) for the February 15[th] 2011 precipitation event at PE. **d)** Same as **c** for the January 13[th] 2015 event at PE. The mean precipitation profile obtained over a long period of observation is also shown and separated into quantiles. The grey dashed lines represent the 20[th] and 80[th] quantiles, the dark dashed line represents the 50[th] quantile and the solid line represents the average of the vertical structure of precipitation (Durán-Alarcón et al., 2019)

This precipitation event is likely generated by the passage of the second low pressure system, as described previously using the corresponding radiosounding. According to Durán-Alarcón et al. (2019), this precipitation event is representative of the climatology of DDU as it lies between the 20th and 80th quantiles (indicated by grey dashed line) with a shape similar to the average climatology in solid black line.

According to figure 3b, there is a poor concordance between the two datasets for low snowfall rate values. The MRR recorded low-level strong values until a null signal of precipitation from 1000 m upward, where CloudSat still records small but significant rates. An inversion of the precipitation rate at low-levels is also observed under the maximum of precipitation rate of 1 mm/h at 600 m. The strong gradient of this inversion is likely due to katabatic wind effects, which can drastically dry out atmospheric layers when blowing down from the ice cap. This event shows that the use of CloudSat for surface precipitation determination may be problematic in certain conditions for a specific event. It is also important to note that this event is an anomalous climatological event in DDU, in comparison with the quantiles of the vertical structure of precipitation both in terms of snowfall rate and shape.

Figure 3c shows a good agreement between the four lowest values of CloudSat observations and the MRR profile. Indeed, every averaged satellite measurement is included in the 40 % confidence interval but the standard deviations indicate a large dispersion. Above this altitude precipitation rate is small and the agreement is weaker. This is similar to what is observed on figure 3b. CloudSat observes again a small signal of precipitation where MRR recorded a null snowfall rate, suggesting some limitations in the sensitivity or attenuation of the MRRs but also a satellite sensitivity for low snowfall rates. This event is an important anomalous climatological event in PE because the observed snowfall rates are much higher than the snowfall rates of Durán-Alarcón et al. (2019) climatology. It is caused by the passage of an atmospheric river over the station.

On figure 3d snowfall rates observed by both CloudSat and MRRs are quite low compared to the three other cases but the agreement remains good for the five satellite lower levels. According to Durán-Alarcón et al. (2019), this precipitation event is representative of the climatology of PE with in particular the presence of virga with very low precipitation rates included between the high and low quantiles.

## 4.2 Agreement between CloudSat and MRR datasets

Figure 4 represents the correlation for (all data, all levels) CloudSat and MRR precipitation reports for the 4 events using the errorbars shown on figure 3. Errorbars for MRRs are implemented by using the confidence intervals obtained with the Ze-Sr relations. Large errorbars correspond to PE's MRR and smaller ones represents the DDU's MRR confidence interval. CloudSat errorbars represent the variance of measurements collected along the swath. A linear regression fit between CloudSat and MRRs is performed, and shows a good correlation between both datasets.

## 4.3 Evidence of a difference between both snowfall rate measurements

A previous study by Protat et al. (2009) showed that CloudSat measured ice cloud reflectivity is 1 dB higher than an airborne cloud radar and a statistical evaluation with basic cloud properties and five ground-based sites showed a weighted-mean difference in Ze which ranges from $-0.4$ dBZ to $+0.3$ dBZ when a period of $\pm 1$ h around the CloudSat overpass is considered.

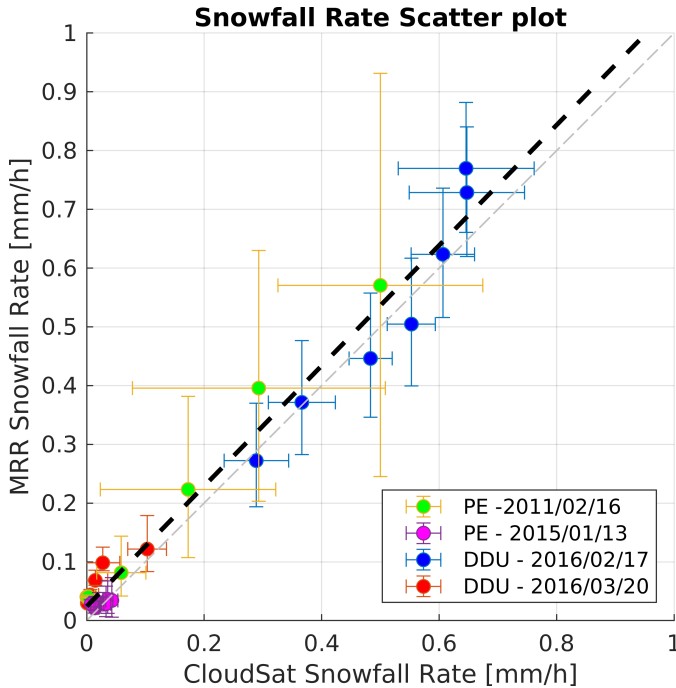

**Figure 4.** Scatter plot of the MRR and CloudSat snowfall rates in mm/h with the linear regression (thick black dash line). The errorbars are computed using the Ze/Sr relations (cf. **Methods.B**) for the MRR and standard deviations at each vertical bin for CloudSat. The grey dashed line represents the 1:1 line for a perfect correlation.

According to Chen et al. (2016), CloudSat tends to observe lighter snowfall events (smaller than 2 mm/h) in comparison with the NOAA/NSSL Multi-Radar Multi-Sensor (MRMS/Q3).

Figure 3b shows that CloudSat can report small but significant snowfall when the MRR signal is virtually zero. The shift between the two instruments is estimated in this case at + 0.040 ± 0.005 mm/h. Then looking at figure 3c for the three last
5   CloudSat bins above 2km height, an averaged snowfall rate of + 0.033 ± 0.003 mm/h is observed when MRR at PE signal is null. Concerning figure 3d, a similar value of + 0.030 ± 0.001 mm/h is recorded by CloudSat, but this time MRR is also recording a similar signal of + 0.029 ± 0.008 mm/h. This difference of measured values suggests a difference in sensitivity of the 2 radars even if these measured rates are above the MRR detection limit of 0.005 mm/h (see section 2.2). This shift in snowfall rates could either be due to a strong attenuation of the MRR backscattered signal with the altitude or to the detection
10   of cloud water by the CPR, as it is more sensitive to small atmospheric particles and clouds.

### 4.4   Calculation of the CloudSat uncertainties

The CloudSat 2C-SNOW-PROFILE product already contains its own uncertainties estimates, calculated from hypothetical parameters such as the mass-diameter distribution of the hydrometeors, their micro-physical and scattering properties. Our analysis suggests that under Antarctic (and probably polar) conditions, this uncertainty can be significantly reduced. By as-

suming that CloudSat and MRR snowfall rates datasets follow a normally-distributed deviation from the mean, a correlation coefficient is calculated in order to establish the degree of similarity between both observations. By using the covariance of both data record, we found a correlation coefficient of 0.99, which confirms a very good agreement between both radar data (see Appendix).

For each CloudSat vertical bin, we calculated the distance of satellite measurement to the corresponding interpolated MRR observation. We averaged these values by weighting them with the MRR confidence intervals and we found a range of CloudSat uncertainties from -13 % up to +22 %.

## 5    Conclusion

CloudSat remote sensing observations were compared with two in-situ Micro-Rain radars at the coastal French Dumont
d'Urville and mountaineous Belgian Princess Elisabeth stations in East Antarctica. The comparison of four cases of precipitation that coincide with CloudSat observations shows a near-perfect correlation. This comparison also reveals a difference in the CloudSat dataset with respect to the MRR for very light precipitation. This might be precipitable cloud water recorded by CloudSat or a MRR limitation due to a strong attenuation of the signal through important precipitation. From our correlation and statistical studies based on the quantification of the CloudSat deviation to the MRR values, we assessed new CloudSat
precipitation uncertainties ranging to -13 % / +22 % based on this short-time and small-space scale study. This new assessment of the CloudSat uncertainties, in spite of the limited number of events, provides confidence in the retrieval given the different climatic and geographical conditions of the two stations. It also justifies further analysis of this dataset in this region of the globe, where snowfall is critical and poorly known. Subsequent studies using weak precipitation rates profiles over other Antarctic regions, particularly in the interior of the continent, will strengthen the robustness of this new range of uncertainties
and corroborate the difference recorded by both CPR and MRRs. Moreover, the EarthCare spaceborne radar, with a much better vertical resolution, should be even more instructive and improve our understanding of clouds and snowfall in the polar regions, where field observations are so hard to perform.

## 6    Appendix

### Calculation of the correlation factor between CloudSat and MRRs

In order to compute the correlation between both datasets, we assume that both the MRRs and CloudSat deviations from the average follow a Gaussian-shaped distribution. MRR data is a Gaussian-shaped distribution, according to its confidence interval calculation. CloudSat deviation from the mean measurements follows also a Gaussian-shaped distribution, as shown on figure 7. Figure 4 shows an evident linear fit between both datasets.

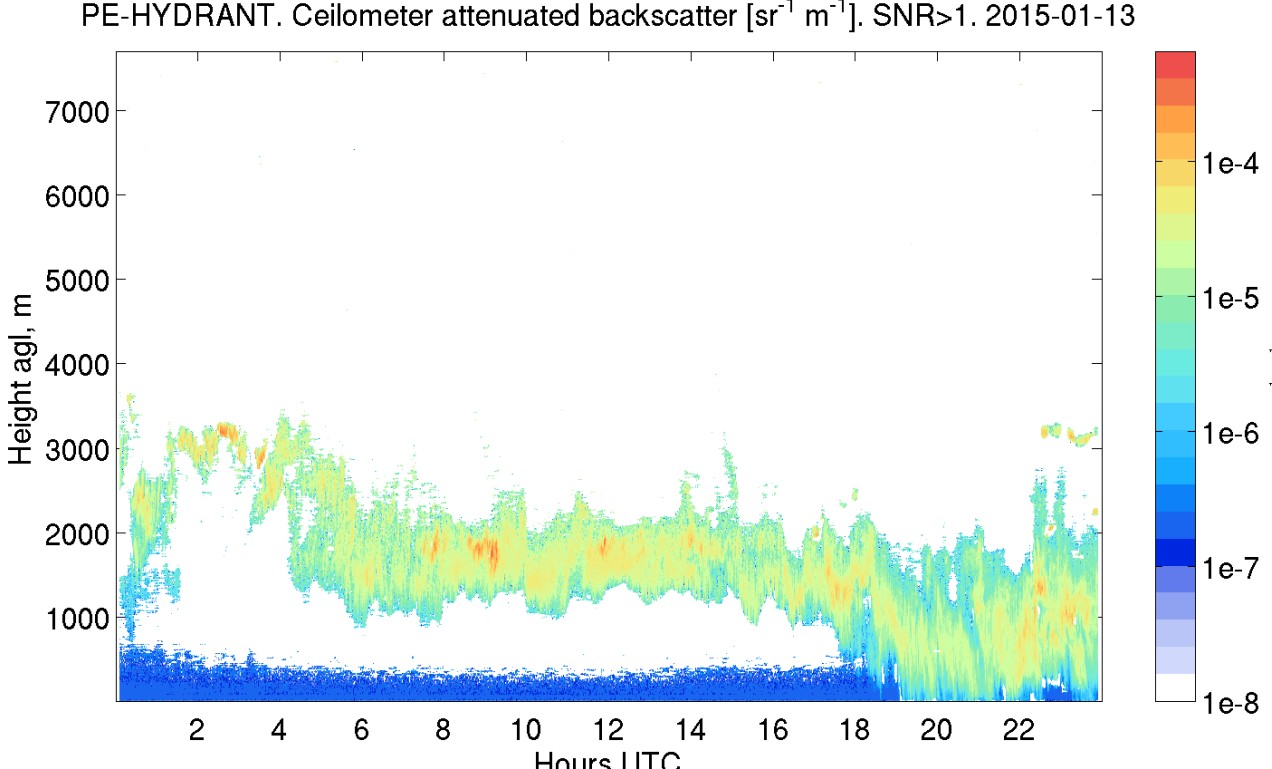

**Figure 5.** Ceilometer backscatter profile at the PE station on January $13^{th}$ 2015. The backscattered reflectivity suggests a passing cloud with in-cloud precipitation and virga.

Because of different vertical bin altitudes, MRR snowfall rate were linearly interpolated at the CloudSat data levels. Covariance of both data populations were calculated by the following equation :

$$cov(S_{CDS}, S_{MRR}) = \frac{\sum_{i=1}^{N}(S_{CDSi} - \overline{S_{CDS}})(S_{MRRi} - \overline{S_{MRR}})}{N} \tag{4}$$

where $S_{CDSi}$ and $S_{MRRi}$ are the snowfall rate values for CloudSat and MRR and $\overline{S_{CDS}}$ and $\overline{S_{MRR}}$ the averaged snowfall rates of both datasets. By calculating the standard deviations $\sigma$ to the mean of each instrument, a covariance matrix were obtained and used to determine the correlation factor $\rho$ between both datasets :

$$\rho = \frac{cov(S_{CDS}, S_{MRR})}{\sqrt{\sigma_{CDS} \, \sigma_{MRR}}} \tag{5}$$

We applied this calculation with both MRR and CloudSat radar datasets and calculated a correlation coefficient of 0.99 as discussed in section 4.2 of Discussions, and showed by a dashed line in figure 4.

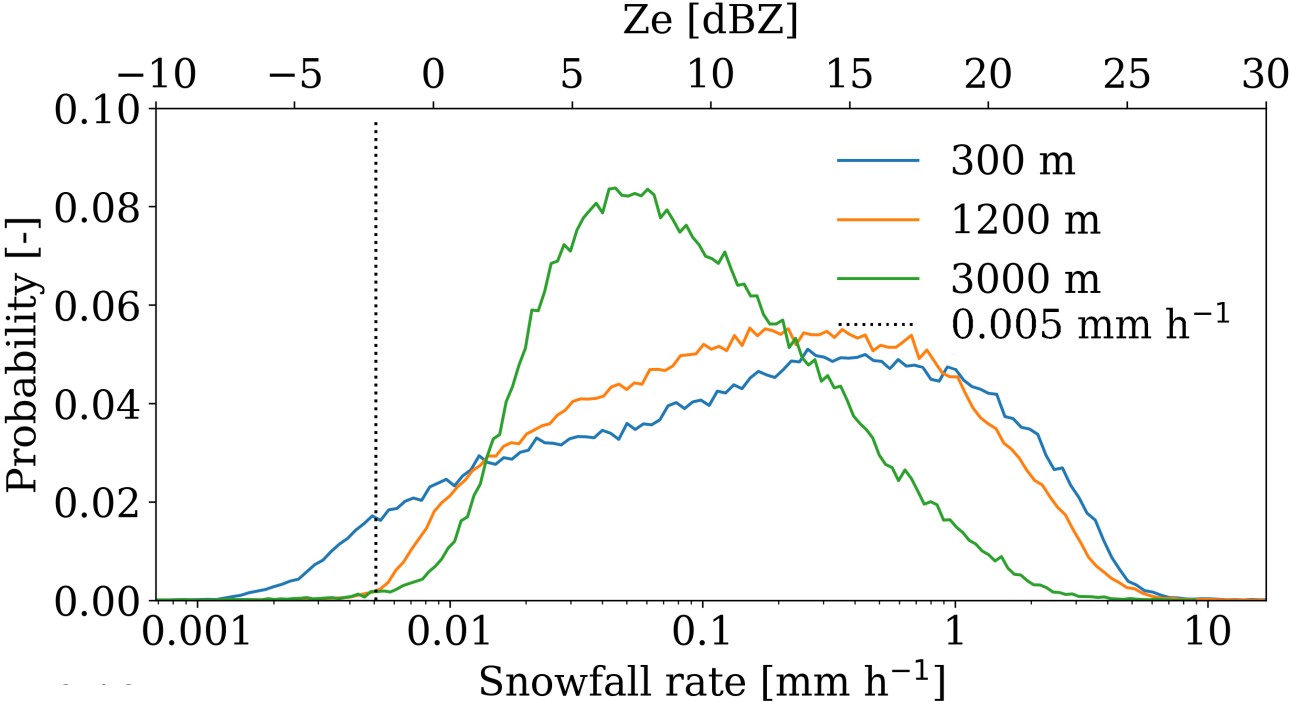

**Figure 6.** Density functions of the corrected 1-minute Ze values at 3 different heights (300m, 1.2km (lowest value of CloudSat) and 3km) at DDU and the respective snowfall rates.

*Competing interests.* The authors declare no conflict of interest.

*Acknowledgements.* Data from the Micro Rain Radar at Dumont d'Urville station have been obtained with the logistical support of the French Polar institute IPEV (program CALVA) and are available at https://doi.pangaea.de/10.1594/PANGAEA.882565. This work was supported by the French National Research Agency (Grant number : ANR-15-CE01-0003). CloudSat data is freely available via the Cloud-Sat Data Processing Center (http://www.cloudsat.cira.colostate.edu/). Data from the Micro Rain Radar at the Princess Elisabeth station can be obtained at http://www.aerocloud.be. This work was supported by the Belgian Science Policy Office (BELSPO; grant number BR/143/A2/AEROCLOUD) and the Research Foundation Flanders (FWO; grant number G0C2215N). I. V. G. thanks the financial support of CESAM (UID/AMB/50017/2019), the FCT/MEC through national funds, and the co-funding by the FEDER, within the PT2020 Partnership Agreement and Compete 2020. The authors thank Jacopo Grazioli for his help and advice during the review period and Anna-Lea Albright as well as Max Popp for their proofreading.

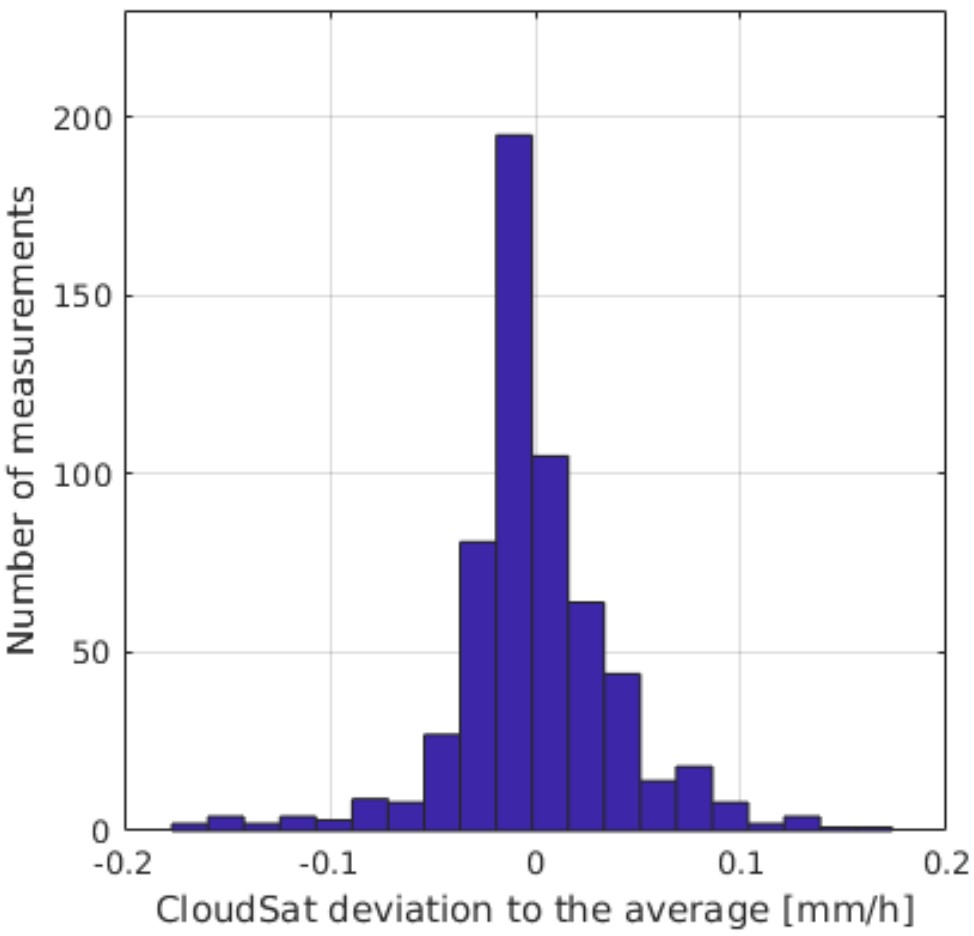

**Figure 7.** Distribution of the deviation from the averaged values of CloudSat snowfall rate for all vertical levels. The deviation from the average is calculated for each considered vertical bin and for each overpass.

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
