# Peer review of "Evaluation of CloudSat snowfall rate profiles by a comparison with in-situ micro rain radar observations in East Antarctica"

_The Cryosphere, 2018_

## Referee Comment (RC1) · Anonymous Referee #1 · 4 Jan 2019

The present work compares and evaluates vertical profiles of CloudSat precipitation product 2C-SNOW-PROFILE with profiles of MRR vertical pointing radars located at Dumont d'Urville and Princess Elisabeth stations in Antarctica. The four considered cases show very good and encouraging results on the comparison (correlation 0.99). This is a really promising result since right now, CPR onboard CloudSat is the only instrument able to give precipitation information over most of Antarctica continent with a reasonable time resolution. The manuscript is very well written, data and methodology are thoroughly described and results are presented in a systematic and logical way. I want to congratulate with the authors because nowadays, as a reviewer, it is very hard to find good work well presented.

[Figure]

I suggest the publication of the manuscript with minor revisions, though I see some points that I would like the authors to clarify.

(1) 4 concurrently snowfall events are considered over 77 actual overflights of CloudSat over the two considered stations. What was the problem of those 77 "events"? probably either MRR or CPR did not detect snow (or both of them). I understand that the detection problem is probably out of the scope of this manuscript, but I would suggest to mention it and explain why those events are not considered, to give to the reader the idea that the problem of snowfall estimate over Antarctica (and in general over the Globe) is not just to quantify it, but we have to deal with detection first of all.

(2) P.5 l.12: the authors provide the Souverijns et al. (2017) Z-S relationship for PE station MRR. As far as I know, the MRR2 have been calibrated with CloudSat, doesn't this introduce a bias in the results of the present work?

(3) P.9 l.26: "in comparison with the quantiles of the vertical structure of precipitation": this should be better explained. I guess you are referring to the black and grey lines in fig.3 that are the 20th, 50th and 80th quantiles and the average precipitation profiles, but also in the figure caption, there is just a reference to Duran-Alarcon et al. I suggest adding some more information both in the text and in the caption to explain better where those plots come from, if they are an average calculated over the station over a certain time period. Moreover, if I am correct, Duran-Alarcon et al. provided reflectivities, how did you get to the snowrates?

(4) P.9 l.31: "also a systematic difference in the CloudSat calibration": this sounds a bit tricky since MRR2 has been calibrated with CloudSat, correct? I guess it is more a sensitivity issue since W- band radars can detect much lighter snowfall than K-band ones.

(5) P.11 l.4: "this precipitation event is representative of the climatology of PE": again, as for p.9 l.18 and l.26 make clear why from the comparison with Duran-Alarcon et al. the event is representative of the climatology (clarify the black and grey lines on the
plots).

(6) P.12 l.5: what do you mean in this case with "higher dispersion"?

(7) P.12 l.22: "by applying to CloudSat profiles the calibration difference estimated in the previous section..": in this case I don't actually understand the procedure you are adopting. You are comparing CloudSat to MRR to evaluate CloudSat, so you are considering MRR as your "truth". But it is known that k-band radar has issues with the detection of light snowrates, so the correction applied doesn't seem to be fair. I would rather look for a minimum detectability threshold for MRR and compare just the rates that both of the sensors are actually able to detect. The comparison of snowfall between different sensors is an hot topic right now and for sure not an easy manageable one, we need to be really careful on the conclusions we take from it.

(8) P.12 sec.4.4: since CloudSat product comes with its own uncertainties, why not consider also them in the analysis and give some advice to the final users of the products that most likely will use that values for their analysis?

Minor comments:

(1) p.3 l.1: use capital H for HYDRological.

(2) p.6 l.5: what do you mean with CloudSat "phase"?

(3) p.6 l.7: "corresponds to a distance": at a first glance this could be confused with the distance from the station, I would suggest adding "covers a distance" or something similar.

(4) p.6 equation: I would suggest using "Vwind" or "Vw" for wind velocity, seems more intuitive.

(5) Fig. 2: in fig. 2g and j include the north direction as you did for the previous two maps.

(6) Fig. 2: here you mention the grey plane disk, in fig.1 was the white disk, be consistent.

(7) Fig. 3: as mentioned on a previous comment, clarify the quantiles information.

(8) Fig. 3: The 80th quantile line in fig.3c became for some reason orange over the shaded orange area instead of gray.

(9) Fig. 4: on the legend use station name and date instead of day number.

(10) Fig. 6: it is not clear from the caption if you are considering each vertical bin of each profile of each overpass (for the 4 considered cases) and then the average value of all of them or if for each overpass and each vertical bin you consider their own average and calculate the deviation from that.

---

## Referee Comment (RC2) · Anonymous Referee #2 · 5 Jan 2019

The authors present a study comparing vertical profiles of snowfall rates obtained from two different radar sensor systems: the CloudSat space-borne platform carrying the Cloud Profiling Radar, and ground-based Micro-Rain Radars (MRRs) installed at two research bases located on the Antarctic continent. A total of four snowfall events are investigated, two for each site where CloudSat overpasses coincided with the operation of the MRRs. Results show good agreement between CloudSat and local MRRs for these four events over the region of ∼960 m to 2500 m. The manuscript is generally very well written and presented.

Specific comments:

[Figure]

1) It appears as though the terms 'dataset' and 'data set' are used interchangeably, please standardise

2) What is 'seconds-short-time'?

3) More explanation is needed about the number of overflights (CloudSat) vs the number of events investigated. I assume that the other overflights that coincided with MRR operation occurred outside of precipitation events? Maybe state this explicitly, the current wording on P3 / 15 was unclear as to whether there was another reason.

4) The 'Methods' sub-section for CloudSat is quite short, it might be useful for the readers if more information such as the revisit time for each station was included. Conspicuously absent is the height AGL of the lowest CloudSat bin used - 1200 m is mentioned in the introduction but it appears as though 960 m is used in Figure 3 a/b but maybe 1050 m is used in c/d.

5) It would be worth adding the 2500 m ceiling used in the MRR data to the MRR method sub-section. It would also be useful to know the spatial extent of the MRR data used for comparison with CloudSat (was it the entire 10 km radius circle used or a subset along the CloudSat track or something else?) Note that it was not abundantly clear what the range of the MRR sensor was, this had to inferred (assuming it was 10 km).

6) On P12 / 8, you already allude ot the fact that these calibrations are different, and the supporting references used elsewhere in this paper (primarily Souverijns et al 2018b) state this. Please clarify the wording here.

7) P12 / 13 It would be very useful to verify whether this is the case, is data on this available? It would also be useful to see whether these values continue further up the CloudSat profile or at other times when the MRR reports 0.

Technical corrections:

P1 / 9 : ', respectively' not needed here

P9 / 9 : 'first lowest' did not make sense to me, maybe pick one?

P12 / 7 : what do you mean by 'higher dispersion'?

Figure 1: Colour scheme of inserts of antarctic contientn make it a bit hard read, it would be better if these stood out more (maybe blue for land mass and/or circular semi-transparent background?)

Figure 2: a/d/g/j inserts of antarctic continent are small and hard to read, given Figure 1 exists these could probably be removed

Figure 3: Altitude often refers to height above MSL, but in this case appears to refer to height AGL, please clarify

---

## Editor Comment (EC1) · Dominé (Editor) · 25 Jan 2019

Dear Authors,

Both reviews are favorable and I clearly encourage you to submit a revised version that addresses the reviewers' comments. As mentioned in my access review, I also suggest checking whether ground-based precipitation or accumulation data are available at DDU and PE, and whether those data can be used as an extra check of CloudSat data. I am fully aware of the limitations of both precipitation gauges and accumulation data, but field scientists will doubtless be interested in the inclusion of such data, if available, in your paper.

[Figure]

Best regards

Florent Domine

---

## Author Comment (AC1) · 22 Feb 2019

Revision of the research article tc-2018-236 Florentin Lemonnier
Laboratoire de Météorologie Dynamique
Couloir 45-55, 3ème étage
75005 Paris, FRANCE

Dear reviewer,

The authors thank you for the review of the manuscript. To clarify our answers to the reviewers comments, the following color scheme is used: comments of the reviewer

are itemized, our answers are denoted in black and quotes from the revised text are in blue. A version of the PDF file showing the differences with the original paper is also included. Please find the answers to the questions you have addressed below.

Sincerely,

Florentin Lemonnier

- 4 concurrently snowfall events are considered over 77 actual overflights of Cloud-Sat over the two considered stations. What was the problem of those 77 "events"? probably either MRR or CPR did not detect snow (or both of them). I understand that the detection problem is probably out of the scope of this manuscript, but I would suggest to mention it and explain why those events are not considered, to give to the reader the idea that the problem of snowfall estimate over Antarctica (and in general over the Globe) is not just to quantify it, but we have to deal with detection first of all.

This is a good point, we thank the reviewer for noting it. As CloudSat overflights over stations occur in a few seconds, it is actually quite unlikely to overpass the stations exactly when precipitation occurs. That is why we see only 4 cases of precipitation out of the 77 overflights. We re-explained it on this study: P3-L15 – With the aim of improving CloudSat radar uncertainty estimates using ground-based observations, CloudSat snowfall retrievals over Dumont d'Urville and Princess Elisabeth stations were compared with MRR data on a total of 4 concurrently recorded snowfall events. During the MRR observing periods, there were 14 overflights over DDU and 63 over PE. These overflights are short, typically a few seconds, explaining why we actually detect snow for only 4 of them.

- P.5 l.: the authors provide the Souverijns et al. (2017) Z-S relationship for PE station MRR. As far as I know, the MRR2 have been calibrated with CloudSat, doesn't this introduce a bias in the results of the present work?

According to Souverijns et al. (2017) authors, the Z-S relationship was performed without CloudSat calibration. Indeed, the authors used a profile comparison for the PE station following the procedure described in Protat et al. (2009; 2010) providing an offset of 1.13dBZ based on profiles of CloudSat within the range of 100 km from the station. This offset has been incorporated in the dBZ values that were used to calculate surface snowfall rates in the Souverijns et al. (2017) study. In our paper, we are using raw MRR data processed with the Maahn and Kollias (2012) algorithm, but not calibrated with CloudSat reflectivities. We added this information in the text : P4-L26 – For this study, the used MRR2 data are processed with the Maahn and Kollias (2012) algorithm. Unlike Souverijns et al. (2017), we did not calibrate the ground radar dataset with CloudSat reflectivities because (1) we want an independent evaluation of the CloudSat CPR dataset, and (2) we do not consider surface precipitation rate comparisons.

- P.9 l.26: "in comparison with the quantiles of the vertical structure of precipitation": this should be better explained. I guess you are referring to the black and grey lines in fig.3 that are the 20th, 50th and 80th quantiles and the average precipitation profiles, but also in the figure caption, there is just a reference to Durán-Alarcón et al. I suggest adding some more information both in the text and in the caption to explain better where those plots come from, if they are an average calculated over the station over a certain time period. Moreover, if I am correct, Durán-Alarcón et al. provided reflectivities, how did you get to the snowrates?

We have inserted more information about the MRRs from Durán-Alarcón's et al., 2019, study in the article: P4-L29 – The mean precipitation profiles obtained over the MRR observation periods (2015-2016 for DDU and 2012 for PE) were also used to evaluate how typical the 4 precipitation events are (Durán-Alarcón et al., 2019). They are obtained using the same Ze/Sr relationships as the ones introduced earlier (see equations (1) and (2)).

- P.9 l.31: "also a systematic difference in the CloudSat calibration": this sounds a bit tricky since MRR2 has been calibrated with CloudSat, correct? I guess it is more a sensitivity issue since W- band radars can detect much lighter snowfall than K-band ones.

We thank the reviewer for this good comment. Indeed, this is more likely due to a difference in sensitivity between the two instruments. We have re-written this in our study: P11-L16 – CloudSat observes again a small signal of precipitation where MRR recorded a null snowfall rate, suggesting some limitations in the sensitivity or attenuation of the MRRs but also a satellite sensitivity for low snowfall rates. In addition, MRR2 is not calibrated with CloudSat for this study as mentioned above: P4-L26 – For this study, the used MRR2 data are processed with the Maahn and Kollias (2012) algorithm. Unlike Souverijns et al. (2017), we did not calibrate the ground radar dataset with CloudSat reflectivities because (1) we want an independent evaluation of the CloudSat CPR dataset, and (2) we do not consider surface precipitation rate comparisons.

- P.11 l.4: "this precipitation event is representative of the climatology of PE": again, as for p.9 l.18 and l.26 make clear why from the comparison with Duran-Alarcon et al. the event is representative of the climatology (clarify the black and grey lines on the plots).

As mentioned above, we compare particular events here with the distribution of all precipitation events recorded by MRRs over the 2015-2016 period, symbolized by quantiles of these distributions (on figure 3, Durán-Alarcón et al., 2019): P4-L29 – The mean precipitation profiles obtained over the MRR observation periods (2015-2016 for DDU and 2012 for PE) were also used to evaluate how typical the 4 precipitation events are

(Durán-Alarcón et al., 2019). They are obtained using the same Ze/Sr relationships as the ones introduced earlier (see equations (1) and (2)).

- P.12 l.5: what do you mean in this case with "higher dispersion"?

Thank you for pointing this oversight, in fact there is no greater dispersion in CloudSat records. This is corrected.

- P.12 l.22: "by applying to CloudSat profiles the calibration difference estimated in the previous section..": in this case I don't actually understand the procedure you are adopting. You are comparing CloudSat to MRR to evaluate CloudSat, so you are considering MRR as your "truth". But it is known that k-band radar has issues with the detection of light snowrates, so the correction applied doesn't seem to be fair. I would rather look for a minimum detectability threshold for MRR and compare just the rates that both of the sensors are actually able to detect. The comparison of snowfall between different sensors is an hot topic right now and for sure not an easy manageable one, we need to be really careful on the conclusions we take from it.

This is a good point and we thank the reviewer for bringing it up. Indeed, what we interpreted as a difference in the calibration of the instruments is more likely to be a difference in the sensitivity of these instruments. We have been considering a MRR detection threshold that we have added hereafter to our study: P6-L2 – According to Maahn and Kollias (2012), the minimum detection of both MRR varies between -14 and -8 dBZ, corresponding to 0.00122 – 0.00546 mm/h at DDU and 0.00385 – 0.0135 mm/h at PE. However these values correspond to theoretical cases of clear sky. Therefore we analyzed the density probability functions of the MRR1 at 3 different levels to determine a minimum threshold of detectability of ground radars (figure 6 in Appendix). We used

the lowest level out of the ground clutter layer (about 1200 m.a.g.l.) and selected a threshold of 0.005 mm/h (see the vertical dashed line in figure 6 in Appendix).

Detection_MRR_DDU2.png

Density functions of the corrected 1-minute Ze values at 3 different heights (300m,

However, although MRRs should be able to do so, we observe that CloudSat detects low snowfall rates below 1 mm/h while ground radar poorly detects them. We propose 2 hypothesis for this difference in snowfall rates : a cloud detection of the CPR or attenuation of the MRR above important low-level precipitation. We added this discussion in our study: P12-L8 – This difference of measured values suggests a difference in sensitivity of the 2 radars even if these measured rates are above the MRR detection limit. This shift in snowfall rates could either be due to a strong attenuation of the MRR backscattered signal with the altitude or to the detection of cloud water by the CPR, as it is more sensitive to small atmospheric particles and clouds.

- P.12 sec.4.4: since CloudSat product comes with its own uncertainties, why not consider also them in the analysis and give some advice to the final users of the products that most likely will use that values for their analysis?

These uncertainties are on instrumental parameters and hypothetical parameters of the hydrometeors. These uncertainties are 1,5 to 2,5 times larger than the measure itself. The aim of our study is to propose a new range of uncertainties estimated in a different and independent way, with ground radars whose range of uncertainty are well known. We have added this information in our study: P12-L12 – The CloudSat 2C-SNOW-PROFILE product already contains its own uncertainties estimates, calculated from hypothetical parameters such as the mass-diameter distribution of the hydrometeors, their micro-physical and scattering properties. Our analysis suggests that under Antarctic (and probably polar) conditions, this uncertainty can be significantly reduced.

- p.3 l.1: use capital H for HYDRological.

This has been corrected.

- p.6 l.5: what do you mean with CloudSat "phase"?

The satellite is characterized by a phase of 16 days, so it exactly overpasses a location every 16 days. We added this information in the CloudSat presentation subsection.

- p.6 l.7: "corresponds to a distance": at a first glance this could be confused with the distance from the station, I would suggest adding "covers a distance" or something similar.

This has been corrected.

- p.6 equation: I would suggest using "Vwind" or "Vw" for wind velocity, seems more intuitive.

This has been corrected.

- Fig. 2: in fig. 2g and j include the north direction as you did for the previous two maps.

This has been corrected.

- Fig. 2: here you mention the grey plane disk, in fig.1 was the white disk, be consistent.

This has been corrected.

- Fig. 3: as mentioned on a previous comment, clarify the quantiles information.

We clarified it in the previous answers.
- Fig. 3: The 80th quantile line in fig.3c became for some reason orange over the shaded orange area instead of gray.

This has been corrected.

- Fig. 4: on the legend use station name and date instead of day number.

This has been corrected.

- Fig. 6: it is not clear from the caption if you are considering each vertical bin of each profile of each overpass (for the 4 considered cases) and then the average value of all of them or if for each overpass and each vertical bin you consider their own average and calculate the deviation from that.

On this figure, the deviation from the average is calculated for each considered vertical bin and for each overpass. We re-explained this in the caption: P16 – Distribution of the deviation from the averaged values of CloudSat snowfall rate for all vertical levels. The deviation from the average is calculated for each considered vertical bin and for each overpass.

---

## Author Comment (AC2) · 22 Feb 2019

Revision of the research article tc-2018-236 Florentin Lemonnier
Laboratoire de Météorologie Dynamique
Couloir 45-55, 3ème étage
75005 Paris, FRANCE

Dear reviewer,

The authors thank you for the review of the manuscript. To clarify our answers to the reviewers comments, the following color scheme is used: comments of the reviewer

are itemized, our answers are denoted in black and quotes from the revised text are in blue. A version of the PDF file showing the differences with the original paper is also included. Please find the answers to the questions you have addressed below.

Sincerely,

Florentin Lemonnier

- It appears as though the terms 'dataset' and 'data set' are used interchangeably, please standardise.

It has been corrected and standardized.

- What is 'seconds-short-time'?

'Seconds-short-time' is used for the satellite because it only observes a weather system by overflying the studied areas in a few seconds. We rewrote that by defining this term: P2-L1 – and short-time (a few seconds) scales

- More explanation is needed about the number of overflights (CloudSat) vs the number of events investigated. I assume that the other overflights that coincided with MRR operation occurred outside of precipitation events? Maybe state this explicitly, the current wording on P3 / 15 was unclear as to whether there was another reason.

Indeed, on all overflights only 4 recorded a precipitation event by both instruments (CPR and MRRs). We explained it: P3-L15 – With the aim of improving CloudSat radar uncertainty estimates using ground-based observations, CloudSat snowfall retrievals

over Dumont d'Urville and Princess Elisabeth stations were compared with MRR data on a total of 4 concurrently recorded snowfall events. During the MRR observing periods, there were 14 overflights over DDU and 63 over PE. These overflights are short, typically a few seconds, explaining why we actually detect snow for only 4 of them.

- The 'Methods' sub-section for CloudSat is quite short, it might be useful for the readers if more information such as the revisit time for each station was included. Conspicuously absent is the height AGL of the lowest CloudSat bin used - 1200 m is mentioned in the introduction but it appears as though 960 m is used in Figure 3 a/b but maybe 1050 m is used in c/d.

Indeed, we mentioned the level at 1200 meters above local surface, but this is an average level for the whole continent firstly used by Palerme et al., 2014. The CloudSat vertical bins are relative to the geoid and depending on the altitude where the stations are located (as shown in the following diagram), the height of the first exploitable bin of CloudSat varies significantly (at PE, the 5th bin is at 1043 m above the surface). Moreover, as we move closer to the ocean, the maximum ground clutter altitude is lower than above the ice cap and the first exploitable bin is generally closer to the surface (at DDU, we are using CloudSat profiles from the 4th bin, which is located at 961 m above the surface). We have also added a paragraph on the characteristics of the satellite, as the phase of the orbits : P3-L32 – The satellite is characterized by a period of 16 days, so it exactly overpasses a location every 16 days. DDU is overpassed by a descending orbit whereas PE is overpassed by ascending and descending orbits which are less than 10 km away. The CloudSat vertical bins are relative to the geoid and depending on the altitude where the stations are located, the first exploitable bin (out of the ground clutter alteration altitude) varies significantly. Moreover, above and near the ocean, when the ice does not interfere much with the radar signal, the ground clutter layer is thinner and lower bins can be used. We are using at DDU CloudSat profiles from the 4th bin, which is located at 961 m.a.g.l. At PE the first exploitable bin

is the 5th, which is located at 1043 m.a.g.l.

```
CPR_DDU_PE.png
```

Schematic diagram of a section of the Antarctic ice cap representing the position of

- It would be worth adding the 2500 m ceiling used in the MRR data to the MRR method sub-section. It would also be useful to know the spatial extent of the MRR data used for comparison with CloudSat (was it the entire 10 km radius circle used or a subset along the CloudSat track or something else?) Note that it was not abundantly clear what the range of the MRR sensor was, this had to inferred (assuming it was 10 km).

We added the maximum altitude of confidence in the MRRs data in this sub-section. This type of radar is characterized by a vertical beam scanning the sky right above him. We have added some technical information on the MRRs, such as the beamwidth: P4-L10 – The MRR is a vertically profiling Doppler radar operating at a frequency of 24.3 GHz (K-band) with a beamwidth of 2° (around 50 m in diameter at 3000 m). At both stations, the resolution was set to 100 m per bin ranging from 300 m – first valid available measurements – to 3000 m. However, we only consider the data up to 2500 m because of the change in the snow microphysical properties above this altitude (Grazioli et al., 2017a).

- On P12 / 8, you already allude ot the fact that these calibrations are different, and the supporting references used elsewhere in this paper (primarily Souverijns et al 2018b) state this. Please clarify the wording here.

We made sure that there is no calibration between the ground radars and CloudSat in our paper. P4-L26 –For this study, the used MRR2 data are processed with the Maahn and Kollias (2012) algorithm. Unlike Souverijns et al. (2017), we did not calibrate the ground radar dataset with CloudSat reflectivities because (1) we want an independent evaluation of the CloudSat CPR dataset, and (2) we do not consider surface precipitation rate comparisons..

- P12 / 13 It would be very useful to verify whether this is the case, is data on this

available? It would also be useful to see whether these values continue further up the CloudSat profile or at other times when the MRR reports 0.

This is a good point and we thank the reviewer for bringing it up. This kind of study has never been done, however we observed on the CloudSat profiles that very light snowfall is recorded up to about 4 km, and in the cases of Fig.3b,c&d when MRR is extinct. Further studies on the CloudSat measurements at higher altitudes would be interesting but this is out of the scope of this paper.

- P1 / 9 : ', respectively' not needed here

This has been corrected: P1-L9 – located in the Dronning Maud Land escarpment zone.

- P9 / 9 : 'first lowest' did not make sense to me, maybe pick one?

This has been corrected, we kept 'first': P9-L20 – Anyway, here MRR measurements are considered and compared to equivalent CloudSat vertical bins only in the first 2500 m of the atmosphere.

- P12 / 7 : what do you mean by 'higher dispersion'?

It was an oversight in the writing, in fact there is no greater dispersion in CloudSat records: P12-L7 – This difference of measured values suggests a difference in sensitivity even if these small measured rates are above the MRR detection limit.

- Figure 1: Colour scheme of inserts of antarctic continent make it a bit hard read, it would be better if these stood out more (maybe blue for land mass and/or circular semi-transparent background?)

This has been improved.

- Figure 2: a/d/g/j inserts of antarctic continent are small and hard to read, given Figure 1 exists these could probably be removed

This has been removed.

- Figure 3: Altitude often refers to height above MSL, but in this case appears to refer to height AGL, please clarify

This has been corrected, all altitudes refers to height AGL.

---

## Author Comment (AC3) · 22 Feb 2019

Revision of the research article tc-2018-236 Florentin Lemonnier
Laboratoire de Météorologie Dynamique
Couloir 45-55, 3ème étage
75005 Paris, FRANCE

Dear editor,

The authors thank you for your detailed editorial work. Based on your advice, we searched for surface precipitation data available for the precipitation events studied in

the article. Unfortunately, no surface data are available at these requested periods. Since then, a PLUVIO2 weighing gauge has however been installed at DDU. The data will be analysed in future papers.

Sincerely,

Florentin Lemonnier

**Supplement:**

[revised manuscript text omitted]